# The Mediating Role of Motivational Regulation on the Relationship of Emotional Intelligence with Physical Activity in Spanish Schoolchildren

**DOI:** 10.3390/children9111656

**Published:** 2022-10-29

**Authors:** Mikel Vaquero-Solís, Miguel Angel Tapia-Serrano, Pedro Antonio Sánchez-Miguel

**Affiliations:** Grupo Análisis Comportamental de la Actividad Física y el Deporte (ACAFYDE), Departamento de Didáctica de la Expresión Musical, Plástica y Corporal, Facultad de Formación del Profesorado, Universidad de Extremadura, 10003 Cáceres, Spain

**Keywords:** adolescents, emotion, motivation physical activity, schools, Self-Determination Theory

## Abstract

The present study aimed to analyze the mediating role of different types of motivational regulations in the relationship established between emotional intelligence and physical activity. Participants were 431 secondary school students aged 12 to 16 years, 51.5% females (13.59 ± 1.03 years) and 48.5% males (13.50 ± 0.94 years), who completed a self-reported questionnaire of emotional intelligence, motivation and physical activity. The results showed a statistically significant positive association between emotional intelligence, physical activity, and more self-determined forms of motivation (intrinsic regulation, identified regulation and introjected regulation) (*p* < 0.05). Several mediation models were also presented that confirmed the mediating value of the more self-determined motivational regulations in the association established between emotional intelligence and physical activity, with the indirect effects being significant for intrinsic regulation, identified regulation, and introjected regulation (*p* < 0.05). Finally, we conclude on the importance of the management of emotions in order to propitiate a suitable motivational state that leads to physical activity. Therefore, this study highlights the importance of emotional intelligence for the practice of different forms of physical activity in young people.

## 1. Introduction

Previous research shows a decrease in physical activity levels during adolescence [1]. In this regard, the performance of physical activity has been associated with multiple benefits, whether physical, such as improved cardiovascular capacity, bone system, and muscular system [2], or psychosocial, such as reduced stress, anxiety, improved management of emotions, and well-being [3]. Therefore, the World Health Organization recommends [4] 60 min of daily moderate vigorous physical activity for children and adolescents. However, in most European countries, less than 50% of children and adolescents do not comply with these recommendations [5]. This fact has driven many researchers whose field of expertise is in the area of school-age health to look for ways to encourage increased healthy behaviors through different approaches and theories, among which those focused on emotion management stand out [6], as do those based on motivation for the initiation and maintenance of behaviors over time [7]. In this respect, important studies developed under the conceptions of emotional intelligence (EI) and motivation have been related to the performance of physical activity [8,9]. However, few studies have used both conceptions simultaneously in relation to physical activity.

Concerning EI, Mayer y Salovey [10] define it as the ability to monitor one’s own and others’ feelings and emotions, discriminate between them, and use this information to guide one’s thinking and actions. To date, there are two main models from which EI has been conceptualized: considering it as a skill [11] or a stable personality trait [12,13]. However, the Laborde’s et al. [14] systematic review proposes the existence of a tripartite model that operates at three levels (knowledge, skill and traits). Likewise, this systematic review indicates that the most widely used model of EI is the trait model. In this sense, the Bar-On´s model [15] is based on identifying traits and skills that help people adapt to the social and emotional demands of life. Thus, the skills referred to Bar On [16] are: (i) intrapersonal intelligence (i.e., the ability to understand emotions in order to be able to express our feelings and communicate with ourselves); (ii) interpersonal intelligence (i.e., the ability to understand our emotions in order to be able to express our feelings and communicate with ourselves); (iii) stress management (i.e., the ability to understand our feelings and communicate with ourselves); (iv) adaptability (i.e., the ability to manage change and solve problems of an intrapersonal and interpersonal nature); and (v) mood (i.e., the ability to generate a positive and self-motivated mood) [13].

On the other hand, motivation has been widely related to the initiation and maintenance of healthy behaviors such as physical activity [7]. In this sense, one of the most widely used motivational theories in the educational context for the performance of physical activity is the Self-Determination Theory (SDT) [9,17]. This theory explains to what extent behaviors are voluntary or self-determined according to the satisfaction or frustration of basic psychological needs (autonomy, competence, and relationships) [18]. Therefore, the more these needs are satisfied, the greater the level of an individual’s motivational regulation within the self-determination continuum. In this regard, Ryan and Deci [19] highlight that motivational behaviors can be divided into three stages (controlled motivation, autonomous motivation, and amotivation). Likewise, these motivational stages are determined by the different forms of motivational regulation that make up the self-determination continuum, which ranges from amotivation to intrinsic regulation. To explain, the continuum of self-determination at one extreme is amotivation or absence of motivation. This is followed by the controlled forms of motivation, namely external regulation, which refers to the performance of an activity to obtain a reward or avoid punishment, and introjected regulation, which refers to the performance of an activity to avoid feelings of guilt or anxiety. Moving to the other end of the continuum (more autonomous forms of motivation), we find identified regulation, which refers to performing an activity that is aligned with the person’s values. Integrated regulation is when the activity itself is aligned with the person’s values and interests. Finally, intrinsic regulation means that the performance of an activity is done for reasons of enjoyment or because it is a challenge for the person [20]. 

With respect to the associations established between EI, motivation, and physical activity, previous studies [21,22] showed that EI is positively associated with more self-determined forms of motivation towards sports practice. Specifically, the systematic review of Ubago-Jiménez et al. [8] concluded that EI is a determining factor for the development of sports competence in the school context. In this sense, a previous study found that more active adolescents presented higher levels of EI [22]. On the other hand, concerning the role that motivation plays in the association between EI and physical activity, the systematic review carried out by Fernández-Espínola and Almagro [23] concluded that the role played by EI in the sequence proposed by the SDT has not yet been analyzed in depth, since it can play a mediating role as well as a predictive role in the different types of motivational regulations. Regarding the predictive role, very few studies have assessed the role of motivation in the association between EI and physical activity in a school population [21,24,25]. In this sense, the work carried out by Arribas-Galarraga et al. [24] and Sukys et al. [21] confirmed the predictive role of EI on different types of motivational regulation in a population of adult athletes. For its part, the work carried out by Porter [25] is the only one that proposes the mediating role of motivation in the relationship between EI and physical activity in a sample of adolescents. However, this work only tests the mediating value of intrinsic regulation, leaving the rest of the regulations that make up the self-determination continuum unassessed. 

Therefore, due to the lack of studies in the school population that analyze the relationships established between emotional intelligence, physical activity, and motivation, the present study aims to analyze in the Spanish context the mediating role of different types of motivational regulation in the relationship between EI and physical activity in a population of adolescent students. Thus, the following hypotheses are derived: Hypothesis 1 (H1), adolescents’ EI will be positively related to physical activity (H1a) and more self-determined levels of motivation (H1b); and hypothesis 2 (H2) the association between EI and physical activity will be mediated by more self-determined levels of motivation. 

## 2. Material and Method

### 2.1. Design and Participants

A cross-sectional design was carried out in four secondary schools in a city in south-western Spain. The schools were selected by simple random sampling. The convenience sample consisted of 431 students aged 12–16 years (13.54 ± 0.99 years), of whom 209 were boys (48.5%; 13.50 ± 0.94 years) and 222 were girls (51.5%; 13.59 ± 1.03 years). In order to participate in the study, participants had to meet a series of inclusion criteria: (i) First, they had to belong to the city of Ceres; (ii) the directors and teachers of the educational centers had to give their consent for the data collection to be carried out; (iii) after obtaining permission from the educational centers, the parents and legal guardians of the participants had to give their consent to participate; (iv) participants had to be in compulsory secondary education at the time of data collection; and (v) finally, the participants must not have participated in a previous data collection that could affect the data collection.

### 2.2. Measures

***Emotional intelligence***. EI was assessed using the Spanish version of the Emotional Quotient Inventory in Young People (Emotional Quotient inventory: Young Version: EQ-i: YV; [26], which has been shown to be valid and reliable for measuring EI in Spanish children and adolescents [27]. This version is composed of 60 items that assess the five dimensions of Bar-On’s EI model. All dimensions showed adequate reliability indices: intrapersonal dimension (6 items, e.g., “I find it easy to tell people how I feel”; α = 0.70), interpersonal (12 items, e.g., “I understand well how other people feel”; α = 0.73), adaptability (10 items, e.g., “It’s easy for me to attend to new things”; α = 0.80), stress management (12 items, e.g., “I can be calm when I am angry”; α = 0.70), and general mood (14 items, e.g., “I am happy”; α = 0.85). In addition, of the 60 items, 6 of them were created by the author to measure the degree to which subjects respond randomly or distort their responses based on social desirability. All responses were answered through a 4-point Likert-type scale, where 1 was very rarely and 4 was very often. Finally, an overall score was made for the EQ variable, which was composed of the mean scores of each of the dimensions.

***Motivation towards physical activity.*** Motivation was assessed using the Spanish version of the Behavioral Regulation in Physical Exercise Questionnaire—2 (BREQ−2); [28,29]. This questionnaire is made up of 19 items grouped into five factors, which refer to the different types of regulation according to the SDT. All items began with the initial sentence “I do exercise...” followed by the factor: intrinsic regulation. (4 items, e.g., “because I think the exercise is fun”; α = 0.84), identified regulation (4 items, e.g., “because I value the benefits of physical exercise”; α = 0.69), introjected regulation (4 items e.g., “because I feel guilty if I don’t practice it”; α = 0.70), extrinsic regulation (4 items e.g., “because others tell me I should do it”, “because others tell me I should do it”; α = 0.70), and amotivation (4 items e.g., “I don’t see why I have to do it”; α = 0.72). All responses were answered on a Likert-type scale from 1 (not true at all) to 5 (completely true). In addition, in this research we used the Self-Determination Index, which was calculated using the following formula: (2 × Intrinsic regulation + Identified regulation) − ((Introjected regulation + External regulation)/2 + 2 × Amotivation) [30]. This index has been shown to be valid and reliable in several studies [31,32]. 

***Physical activity***. The physical activity measure was calculated using the Spanish-validated Physical Activity Questionnaire for Adolescents (Physical Activity Questionary for Adolescents: PAQ-A) [33]. This questionnaire is composed of nine items that assess the level of physical activity that the adolescent performed in the last 7 days during physical education classes, leisure time, or different time segments throughout the day (lunch, afternoons, and evenings). The score obtained can vary from 1 to 5, allowing a graduation in the level of physical activity to be established. All responses were answered on a Likert-type scale from 1 (none or not at all) to 5 (5 times or more or a lot). Finally, the Cronbach’s alpha coefficient obtained for the present sample was (α = 0.79).

### 2.3. Procedore

This study was conducted in accordance with the ethical principles of the Declaration of Helsinki and the Ethics Committee of the university to which the authors are members (26/2020). Likewise, all participants have been treated according to the ethical principles of the American Psychological Association, respecting the freedom of participation, confidentiality and anonymity of the responses. Regarding the data collection process, this took place in the pre-pandemic period located in the middle of 2017. In this sense, the research team contacted the principals and teachers of the schools to assess their degree of agreement in participating in the project. Once the participation of the management team and teachers was achieved, a letter of consent was sent to fathers, mothers, and legal guardians, which they had to bring signed if they wished to participate in the project. For data collection, a member of the research team was in the classroom in case any questions arose. The time dedicated to completing the questionnaire was around 20–25 min.

### 2.4. Data Analyses

Data analyses was performed using the IBM SPSS 23.0 (New York, NY, USA) statistical package. First, different normality tests were performed to determine the nature of the data (Kolmogórov-Smirnov, and Rachas Test). The results obtained revealed the parametric characteristics of the data. Subsequently, descriptive statistics and correlation analysis were performed for all the variables in the study. For the analysis of the mediating effect of the different types of motivational regulation on the association between the EI and physical activity, the bootstrapping method was applied through the PROCESS SPSS extension of the Hayes macro version 4.00 [34]. In the PROCESS extension, model 4 (recommended for mediation analysis with a mediator) was established with 10,000 interactions to determine the mediation of the six regression models (intrinsic regulation, identified, introjected, external, amotivation, and self-determination index). Mediation analyses reported the unstandardized (B) and standardized (β) regression coefficients, as well as the percentage of the total effect. The regression coefficient of EI on the different types of regulation was denoted as path *a*. Secondly, the regression coefficient of EI on physical activity considered as a total effect was denoted as equation *c*. Third, the regression coefficients of both EI and the different types of regression in relation to physical activity were composed through equations *b* and *c* and referred to as the direct effect. Finally, to test for indirect effects, a bootstrapping resampling of 10,000 was performed for the calculation of 95% confidence intervals. Statistical significance was considered for *p* < 0.05.

## 3. Results

### 3.1. Descriptive Statistics

Table 1 shows the descriptive statistics and correlation analyses of all the variables in the study. Boys showed higher levels for EI, physical activity, and in all motivational regulations, except for the self-determination index, which was higher for girls. Likewise, correlation analyses showed significant positive correlations between EI total score, physical activity, and intrinsic, integrated, and introjected regulations of motivation (*p* < 0.05).

### 3.2. Main Analysis

Figure 1 shows the mediating role for each of the levels of motivational regulation in the associations between EI and physical activity. The results of the various mediation analyses showed significant indirect effects for intrinsic regulation (*β* = 0.08; 95% CI 0.03, 0.12), identified regulation (*β* = 0.12; 95% CI 0.07, 0.17), introjected regulation (*β* = 0.03; 95% CI 0.00, 0.06), and self-determination index (*β* = 0.07; 95% CI 0.03, 0.11) on the associations produced between EI and physical activity. All of them showed significant indirect effects on physical activity (*p* < 0.05). Therefore, the level of EI affects the amount of physical activity performed through the different motivational regulations possessed by the participants. In contrast, external regulation and demotivation did not show significant indirect effects.

## 4. Discussion

The present study analyzed the mediating role of the different types of motivational regulation according to SDT in the relationship between EI and physical activity. The main results of this study showed a significant mediating value of the most self-determined forms of motivational regulation for the association between EI and physical activity.

Regarding the first hypothesis, correlation analyses confirmed significant positive associations between EI and physical activity. These results are consistent with those found in previous studies [8,14]. In this sense, there is strong evidence on the role that some personality traits play in the performance of physical activity [6]. As for the second part of the first hypothesis, it was stated that EI would be positively related to the most self-determined forms of motivational regulation. Our results showed significant positive associations of EI with the most self-determined forms of motivation. These findings are congruent with those found in [22,35], which could be explained by the role of EI in motivation. In this sense, Baumeister et al. [36] point out that depending on a person’s personal motivations, they will highlight what is relevant to the individual through emotional responses.

The second hypothesis stated that the association between physical activity and EI will be mediated by the most self-determined types of regulation. In this sense, concerning the most self-determined types of motivation within the self-determination continuum, our results confirmed the mediating value for intrinsic and identified regulation. These findings are in line with those found in the work of [25,37] and [21], where the predictive role of EI in the self-determination continuum is also shown. Specifically, the work of Sukys et al. [21] showed how general EI predicted the more autonomous forms of motivation (intrinsic, integrated and identified regulation). This fact could be explained through two internal processes: on the one hand, through the role played by emotions in the self-motivation for the performance of physical activity [37], and on the other hand, the role of the most self-determined forms of motivation in the performance of physical activity. In this sense, the performance of physical activity can be explained by the emotions or interests that a person presents and by the valuation or challenge that the person himself/herself makes about the activity to be performed. This explanation is in line with the postulates presented by the SDT for intrinsic and identified regulation [18]. Likewise, these findings are hardly debatable, since few studies have assessed the predictive role played by EI in the SDT sequence. However, Roth et al. [38] have recently attempted to give an integrative approach to SDT by incorporating emotions that support autonomous behaviors. Thus, being emotionally aware of aspects of oneself (short-term, long-term goals, values or preferences) implies making better decisions with respect to subsequent actions [39,40]. Therefore, it can be deduced that emotionally more intelligent individuals are more likely to be intrinsically more motivated. 

On the other hand, in the series of mediation analyses performed, our results also showed that for introjected regulation, one of the least self-determined forms of motivation presented a significant mediating value. This fact is consistent with the study of [21], where EI was significantly positively associated with introjected regulation. Therefore, we could interpret that emotions generated by feelings of self-blame, shame, anxiety, or ego may mediate the associations produced between EI and physical activity. In this regard, Sukys et al. [21] expose that people with a good level of EI see problems as learning situations, handling stressful situations better and improving their motivation in the practice of physical activity. In the same way, Gillison et al. [41] point out that the motivational regulation derived from a feeling of guilt may involve the management of emotions towards the performance of a certain activity (e.g., physical activity).

The results of this study have shown how EI, understood as the ability to monitor one’s feelings and emotions, influence behaviors (i.e., performance of physical activity). However, we should take these findings with caution, as we are still immersed in the debate on “how behaviors are produced” (emotion or motivation or a mixture of both traits) [36].

### Limitations and Strengths

This research shows some limitations, such as the design used in the study, which does not allow us to extract cause–effect relationships. In this regard, future longitudinal studies are needed to confirm the associations analyzed in this study. Likewise, we should mention that, although valid and reliable questionnaires were used in this study, these were self-reported, so future studies should make use of objective devices, such as accelerometers, for a more detailed recording of physical activity. Moreover, future work should test how motivational constructs moderate the relationship between EI and physical activity. Thus, a set of traits could be identified within the participants that affect greater physical activity performance. Finally, the small sample size may pose a drawback when extrapolating these results to other populations.

On the other hand, the study also shows important strengths that should be highlighted, such as the novelty of the study and the bibliographic contributions made. In this sense, the study contributes to respond to a research gap due to the absence of studies that have proven the mediating value of motivation for the relationships produced between EI and physical activity. Similarly, another strength to highlight is the population of adolescents used. In this case, most studies have assessed these relationships in athletes or sportsmen. Finally, the main strength of the study is that, to our knowledge, it is the first to assess the mediating role of different types of motivational regulation in the association between EI and physical activity.

## 5. Conclusions

The present study showed the mediating role of the more self-determined types of SDT regulations (intrinsic regulation and identified regulation) in the relationship between EI and physical activity. Likewise, within the less self-determined forms of SDT, the value of introjected regulation was striking. For their part, external regulation and demotivation did not affect the established association between EI and physical activity. Therefore, it seems that adolescents who better handle and manage their emotions are more likely to present a more adequate motivational state that has an impact on physical activity performance. In the same way, participants who have a more self-determined type of motivational regulation will find it easier to manage feelings or emotions that lead them to physical activity or other behaviors. Finally, participants with a more self-determined type of motivational regulation are more likely to have more adequate levels of motivation, which has an impact on the initiation and adherence of behaviors.

## Figures and Tables

**Figure 1 children-09-01656-f001:**
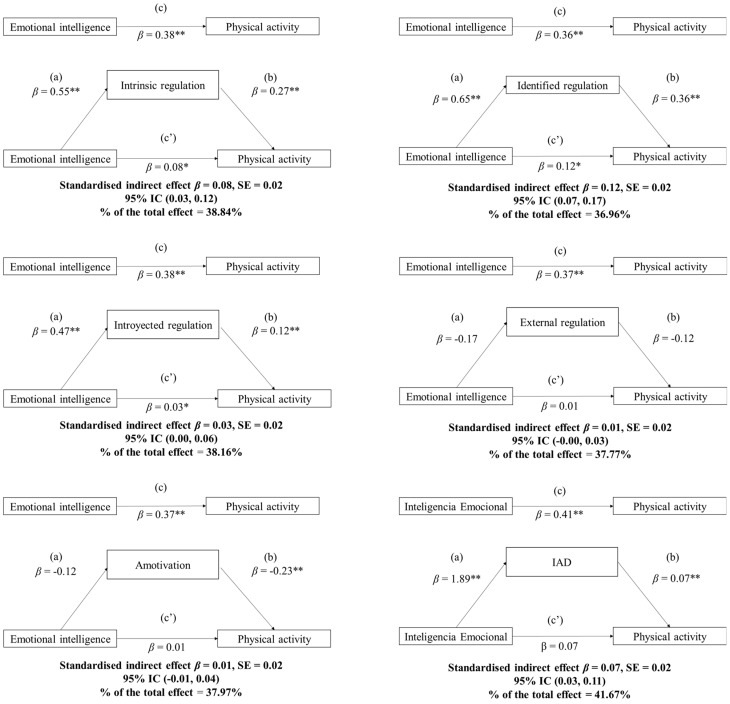
Results of the mediation analysis on the different types of motivational regulation on the association between emotional intelligence and physical activity. Note: ** *p* < 0.01; * *p* < 0.05; Path *a* = The regression coefficient of EI on the different types of motivational regulation; *Path b* = The regression coefficient of different types of motivational regulation on physical activity; *Path c* = The regression coefficient of EI on physical activity considered as a total effect. *Paht c’ =* the regression coefficients of both EI and the different types of regression in relation to physical activity were composed through equations b and c and referred to as the direct effect.

**Table 1 children-09-01656-t001:** Descriptive analysis and bivariate correlations.

	Boys		Girls		1	2	3	4	5	6	7	8
	M	SD	M	SD								
1. EI	3.44	0.34	3.41	3.50	-	0.22 **	0.29 **	0.14 **	−0.08	−0.06	0.20 **	0.20 **
2. Intrinsic Regulation	4.08	0.90	3.93	0.95		-	0.59 **	0.11 *	−0.27 **	−0.47 **	0.87 **	0.42 **
3. Identified Regulation	3.60	0.73	3.50	0.81			-	0.47 **	−0.05	−0.31 **	0.61 **	0.44 **
4. Introjected Regulation	2.58	1.07	2.49	1.05				-	0.24 **	−0.03	0.00	0.23 **
5. External Regulation	1.80	0.83	1.67	0.74					-	0.43 **	−0.49 **	−0.11 *
6. Amotivation	1.73	0.84	1.53	0.67						-	−0.79 **	−0.22 **
7. SDI	6.16	3.57	6.21	3.42							-	0.40 **
8. Physical activity	2.89	0.68	2.70	0.65								-

**Note.** ** *p* < 0.01; * *p* < 0.05; M = Means; SD = Standard Deviation; EI = Emotional Intelligence; SDI = Self-Determination Index.

## Data Availability

The datasets used and/or analyzed during the current study are available from the corresponding author on reasonable request.

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
