# Peer review of "The Mediating Role of Motivational Regulation on the Relationship of Emotional Intelligence with Physical Activity in Spanish Schoolchildren"

_children, 2022, doi:10.3390/children9111656_

Round 1
Reviewer 1 Report
The mediating role of motivational regulation on the relationship of emotional intelligence with physical activity in Spanish schoolchildren
General Comment: Whereas the topic is of interest to the field, two weaknesses in the paper are currently evident:
1- The paper is in significant need of English language editing by someone in the field- the paper lacks clarity in intent and discussion of results.
Is the paper more about the role of emotional intelligence (EI)? As noted in the final statement of the Abstract?
“Therefore, this study highlights the importance of emotional intelligence for the practice of sports in young people.”
Or, motivational regulation and EI? And, its role in physical activity. This is not clear.
And,
2- Related to the clarity of language usage is lack of specificity of the results.
“Results showed a statistically significant positive association between emotional intelligence, physical activity and more self-determined forms of motivation.” More specifically, what characteristics of EI and self-determined forms of motivation?
The researchers describe physical activity and sports, e.g., in the last sentence of the Abstract “…. for the practice of sports in young people.” You were measuring physical exercise… the samples activity could be a number of physical activities and not sports per se.
Specific Comments:
Introduction
“This fact has conducted many school-age health researchers to look for ways to encourage the increase of healthy behaviors through different approaches and theories, among which we can highlight the research focused on the management of emotions.” Awkward use of English
I would use past tense in sentences such as “… Fernández-Espínola and Almagro [23] concludes …” use concluded; there are several of these.
Use the abbreviation EI for emotional intelligence after first use.
Overall, the review of EI and motivation theory as related to the topic is good.
“Therefore, there is a lack of studies in the school population that study the relationships established between emotional intelligence, physical activity and motivation.” Not established yet… that is your aim… again, English usage.
Again, “….that proposes to analyze the mediating role of different types of motivational regulation in the relationship between intelligence and physical activity in a population of adolescent students.” No, this is the first to analyze…or examine.
Method
Measures are good.
Results/Discussion
Figure 1. Association emotional intiligence. ??? Also, there is much too much in this figure. State in text and consolidate results if possible, or highlight the most significant.
“The present study tested the mediating role of the different motivational constructs of the SDT in the relationship between EI and physical activity.” The SDT? This does not align very well with introductory intent. Be specific to aims.
Hypothesis link is good.
“The results of this study have shown how emotional intelligence, understood as the ability to monitor one's feelings and emotions, influence behaviors (i.e., performance of physical activity).” Again, as noted in the remarks on the abstract and introduction, is this study more of the influence of EI? As noted here, or on other?
Conclusion
“The present study showed the mediating role of the more self-determined types of regulations of the SDT in the relationship between emotional intelligence and physical activity.” What is missing is clarity of the specific role of more self-determined types and EI. Your results and abstract appear to focus of EI and not the relationship with SDT.
I believe that the results are there, but the writing of this paper makes it very confusing and unclear.
Author Response
The mediating role of motivational regulation on the relationship of emotional intelligence with physical activity in Spanish schoolchildren
Reviewer#1
The mediating role of motivational regulation on the relationship of emotional intelligence with physical activity in Spanish schoolchildren
General Comment: Whereas the topic is of interest to the field, two weaknesses in the paper are currently evident:
We would like to thank the editor and reviewers for the opportunity to respond to their suggestions for improvement. Below are the point-by-point responses to the reviewers' thoughtful comments. Thank you very much. In this document, we indicate the modifications made in the new version of the manuscript, in addition to specifying the page(s) where these modifications can be found.
Question 1: The paper is in significant need of English language editing by someone in the field- the paper lacks clarity in intent and discussion of results.
Answer 1: Thanks to reviewer for his comments. In this regard, the authors have proceeded to send the article to a translation service performed by native speakers.
Question 2: Is the paper more about the role of emotional intelligence (EI)? As noted in the final statement of the Abstract?
“Therefore, this study highlights the importance of emotional intelligence for the practice of sports in young people.”
Or, motivational regulation and EI? And, its role in physical activity. This is not clear.
Answer 2: Thanks to reviewer for his interesting question. In this sense, when designing the research, the authors asked themselves: what drives the behavior of young schoolchildren, motivation or emotion?
This question led to a process of reviewing the scientific literature, reaching the conclusion that behaviors might be governed, at least in part, by emotional intelligence (Laborde et al., 2016, p 862). therefore, we decided to highlight the role of emotional intelligence in the performance of physical activity mediated by different types of motivational regulations (i.e., we highlighted the role of emotional intelligence for the practice of physical activity without forgetting the construct of motivation).
Laborde, S.; Dosseville, F.; Allen, M.S. Emotional intelligence in sport and exercise: A systematic review. Scand. J. Med. Sci. Sport. 2016, 26, 862–874, doi:10.1111/sms.12510.
Question 3: Related to the clarity of language usage is lack of specificity of the results.
“Results showed a statistically significant positive association between emotional intelligence, physical activity and more self-determined forms of motivation.” More specifically, what characteristics of EI and self-determined forms of motivation?
Answer 3a: we welcome suggestions for improvement. In this regard, the authors have considered emotional intelligence as a global score, as previous research has also used the overall construct for their investigations (Bechter et al., 2021; Sukys et al., 2019; González-Valero et al., 2019). Likewise, with respect to the forms of motivation, we have specified the types of regulation whose mediation analyses were significant. Therefore, we have added in the abstract text the follow information:
The results showed a statistically significant positive association between emotional intelligence, physical activity and the most self-determined forms of motivation (intrinsic regulation, identified regulation and introjected regulation) (page, 1. Lines 17-18)
Sukys, S., TilindienÄ—, I., Cesnaitiene, V. J., & Kreivyte, R. (2019). Does emotional intelligence predict athletes’ motivation to participate in sports?. Perceptual and motor skills, 126(2), 305-322.
González-Valero, G., Zurita-Ortega, F., Chacón-Cuberos, R., & Puertas-Molero, P. (2019). Analysis of motivational Climate, Emotional Intelligence, and Healthy Habits in Physical Education Teachers of the future using structural equations. Sustainability, 11(13), 3740.
Bechter, B. E., Whipp, P. R., Dimmock, J. A., & Jackson, B. (2021). Emotional intelligence and interpersonal relationship quality as predictors of high school physical education teachers’ intrinsic motivation. Current Psychology, 1-9.
The researchers describe physical activity and sports, e.g., in the last sentence of the Abstract “…. for the practice of sports in young people.” You were measuring physical exercise… the samples activity could be a number of physical activities and not sports per se.
Answer 3b: We would like to thank the reviewer for his comment. In this regard, we agree with the nomenclature error about physical activity/sport/physical exercise proposed by the reviewer. therefore, we have modified the last sentence of the abstract, since the sporting practice we refer to is the practice of physical activity that is neither regulated nor systematic.
Thus, we have added the following information in the text: “the importance of emotional intelligence for the practice of different forms of physical activity in young people.” (Page 1, line 23)
Specific Comments:
- Introduction
Question 4: “This fact has conducted many school-age health researchers to look for ways to encourage the increase of healthy behaviors through different approaches and theories, among which we can highlight the research focused on the management of emotions.” Awkward use of English
Answer 4: Thanks to the reviewer for his comments. We regret the Awkward use of English, which we have tried to solve with a new sentence restructuring: This fact has driven many researchers, whose field of expertise is in the area of school-age health, to look for ways to encourage increased healthy behaviors through different approaches and theories, among which those focused on emotion management stand out (page 1, lines 34-36)
Question 5: I would use past tense in sentences such as “… Fernández-Espínola and Almagro [23] concludes …” use concluded; there are several of these.
Answer 5: We thank to the reviewer for his opinion to improve the quality of the manuscript. In this regard, we have changed several sentences by modifying the past tense in the sentences:
- Showed (page 2, line 82)
- Conluded (page 2, line 84 and 89)
- Confirmed (page 2, line 95)
Question 6: Use the abbreviation EI for emotional intelligence after first use.
Answer 6: We thank for the reviewer's recommendation. Therefore, we have replaced emotional intelligence with its abbreviation throughout the manuscript (page 1, line 39) first use
Question 7: Overall, the review of EI and motivation theory as related to the topic is good.
Answer 7: We would like to thank the reviewer for his opinion on the review of EI and motivation. We would like to thank the reviewer again for his favorable comments.
Question 8: “Therefore, there is a lack of studies in the school population that study the relationships established between emotional intelligence, physical activity and motivation.” Not established yet… that is your aim… again, English usage.
Answer 8: Thanks to the reviewer for his comment. In this regard, the authors have modified the following sentence to make it clearer and more understandable:
- Therefore, due to the lack of studies in the school population that analyze the relationships established between emotional intelligence, physical activity and motivation, the present study aims to analyze in the Spanish context the mediating role of different types of motivational regulation in the relationship between EI and physical activity in a population of adolescent students. (Page 3, lines 99-103)
Question 9: Again, “…. that proposes to analyze the mediating role of different types of motivational regulation in the relationship between intelligence and physical activity in a population of adolescent students.” No, this is the first to analyze…or examine.
Answer 9: Thank you for your comment we have tried to unifying terms to “analyze” (page 3, lines 99 and 101)
- Method
Measures are good.
Answer: We would like to thank the reviewer again for his favorable comments.
- Results/Discussion
Question 10: Figure 1. Association emotional intiligence. ??? Also, there is much too much in this figure. State in text and consolidate results if possible, or highlight the most significant.
Answer 10: We appreciate the reviewer's suggestions to improve the quality of the manuscript. In this regard, the authors have chosen to simplify the information in Figure 1, highlight the most significant and consolidate the results. Therefore, we added the following information: The results of the various mediation analyses showed significant indirect effects for intrinsic regulation (β = 0.08; 95% CI 0.03, 0.12), identified regulation (β = 0.12; 95% CI 0.07, 0.17), introjected regulation (β = 0.03; 95% CI 0.00, 0.06) and self-determination index (β = 0.07; 95% CI 0.03, 0.11) on the associations produced between EI and physical activity (page 4-5, lines 184-188). In addition, we have chosen to change the title of Figure 1 to be more in line with the results presented. Thus, the title of Figure 1 is now “Results of the mediation analysis on the different types of motivational regulation on the association between emotional intelligence and physical activity” (page 5, lines 193-194)
Question 11: “The present study tested the mediating role of the different motivational constructs of the SDT in the relationship between EI and physical activity.” The SDT? This does not align very well with introductory intent. Be specific to aims.
Answer 11: Thank to the reviewer for his comment. In this regard, the authors the authors have changed the wording so as not to cause inferences between the introduction and the discussion.
The sentence has been reworded as follows: The present study analyzed the mediating role of the different types of motivational regulation according to SDT in the relationship between EI and physical activity (page 6, lines 197-198)
Thus, the verb of the objective now agrees with the one stated in the introduction and the word "motivational construct" has been changed to "motivational regulations".
Hypothesis link is good.
Question 12: The results of this study have shown how emotional intelligence, understood as the ability to monitor one's feelings and emotions, influence behaviors (i.e., performance of physical activity).” Again, as noted in the remarks on the abstract and introduction, is this study more of the influence of EI? As noted here, or on other?
Answer 12: Thanks to the reviewer for his interesting question. In this regard, kindly, at the beginning of this response letter we have tried to answer this interesting question and clear your doubts. Similarly, we will try to supplement the answer. Thus, to provide an answer to this question, we must focus on the beginning of the research and its design, by which we propose that emotional intelligence is the independent variable that influences physical activity through the most self-determined motivational regulations. Thus, this manuscript focuses on IE but takes into account the motivation
Conclusion
Question 13: “The present study showed the mediating role of the more self-determined types of regulations of the SDT in the relationship between emotional intelligence and physical activity.” What is missing is clarity of the specific role of more self-determined types and EI. Your results and abstract appear to focus of EI and not the relationship with SDT.
I believe that the results are there, but the writing of this paper makes it very confusing and unclear.
Answer 13: Thank to the reviewer for his comment. In this regard, the authors have tried to provide more clarity in the text by complementing the explanation with the role of motivational regulation through the association between EI and physical activity. Thus, we have added a few more lines. (Page 8, lines 294, 300-301). In addition, we have submitted the manuscript to a proof-reading service to fix the English.

Reviewer 2 Report
Title: The mediating role of motivational regulation on the relationship of emotional intelligence with physical activity in Spanish schoolchildren
Article Type: Article
Summary
In this article, the authors aimed to analyze the mediating role of different types of motivational regulations in the relationship established between emotional intelligence and physical activity among 431 secondary school students aged 12 to 16 years. The emotional intelligence, motivation and physical activity were assessed by self-reported questionnaire. The results indicated that there is a positive relationship between emotional intelligence, physical activity and motivation. The results also indicated a mediating role of the motivational regulations in relationship between emotional intelligence and physical activity.
Evaluation
The topic of this study is interesting for publication in the Journal. The sample size and the design for the study is appropriate to answer the research questions, and the paper is well written. However, there are some points should be addressed by the authors, in order to improve the quality of the manuscript.
Points and suggestions
This sentence is replicated at the abstract, please remove it.” Results showed a statistically significant positive association between emotional intelligence, physical activity and more self-determined forms of motivation.”
Please speak in detail about the results in the abstract.
What was the reason for choosing the participants in this age group? Is it just because of the lack of research at this age in Spain? It is suggested to mention better reasons for using this age group in the introduction.
Please add the inclusion and exclusion to the study in the method section
How is the sample size is calculated?
What date was the research done? Was it in the middle of the COVID-19 pandemic? Please explain more about this in method section.
Please speak in detail about the Hayes macro method in the method section.
Wouldn't it be better if you used objective methods of measuring physical activity? It is suggested to be mentioned in the limitations section.
Please search the research literature more carefully. It seems that similar research has been done recently.
Author Response
The mediating role of motivational regulation on the relationship of emotional intelligence with physical activity in Spanish schoolchildren
Reviewer#2
Summary
In this article, the authors aimed to analyze the mediating role of different types of motivational regulations in the relationship established between emotional intelligence and physical activity among 431 secondary school students aged 12 to 16 years. The emotional intelligence, motivation and physical activity were assessed by self-reported questionnaire. The results indicated that there is a positive relationship between emotional intelligence, physical activity and motivation. The results also indicated a mediating role of the motivational regulations in relationship between emotional intelligence and physical activity.
Evaluation
The topic of this study is interesting for publication in the Journal. The sample size and the design for the study is appropriate to answer the research questions, and the paper is well written. However, there are some points should be addressed by the authors, in order to improve the quality of the manuscript.
Points and suggestions
Question 1: This sentence is replicated at the abstract, please remove it.” Results showed a statistically significant positive association between emotional intelligence, physical activity and more self-determined forms of motivation.”
Answer 1: Thank to the reviewer for his comment. In this regard, we the authors have followed the reviewer's suggestion and removed the replicated sentence in the abstract. (Page 1, line 16-17)
Question 2: Please speak in detail about the results in the abstract.
Answer 2: We thank to the reviewer for his opinion to improve the quality of the manuscript. In this regard, we have added more information to give more detail about the results in the abstract. (Page 1, line 17, 20-21)
Question 3: What was the reason for choosing the participants in this age group? Is it just because of the lack of research at this age in Spain? It is suggested to mention better reasons for using this age group in the introduction.
Answer 3: Thanks to reviewer for his interesting question. In this sense, the authors made an extensive review of the studies that had worked jointly on motivation and emotional intelligence. In this regard, we found that motivation and emotional intelligence had not been a much studied topic. This fact led us to initiate the work under the premise that the ability to monitor our own feelings and emotions (EI) is mediated by different motivational regulations in association with physical activity. Subsequently, we proceeded to review several papers that had analyzed EI with physical activity, the most significant being (Laborde et al., 2016).
All this led us to identify several research gaps: the first was the absence of studies assessing the mediation capacity of motivation in the association between IE and physical activity. Secondly, after reading the systematic reviews, we saw that few studies had focused on the school population. There was only a total of 4 studies in children and adolescents. Likewise, most of the studies were focused on athletes, university students and adults. Finally, in the Spanish context, there were no studies that analyzed the mediating role of motivation in the association of IE with physical activity.
Therefore, the authors point out in the text that very few studies have evaluated the role of motivation in the association of EI with physical activity in the adolescent population and, likewise, we also point out the absence of studies in the school population (page 2-3, lines 98-101).
Question 4: Please add the inclusion and exclusion to the study in the method section
Answer 4: Thanks to the reviewer for his comment. The inclusion criteria has been added in the section method (Page 3, lines-114-121)
Question 5: How is the sample size is calculated?
Answer 5: Thanks to the reviewer for his comment. The sample size was calculated using the Gpower 3.1.9.2 statistical package. In this sense, the input parameters determined according to the statistical tests to which they were subjected were completed. Therefore, appropriate statistical tests were selected and performed a priori (prior to the assessment of the actual data) with a regression model with two predictor variables and a three percent experimental deadweight. In this sense, the program gave us an n = 107 participants with a confidence interval of 95%.
Question 6: What date was the research done? Was it in the middle of the COVID-19 pandemic? Please explain more about this in method section.
Answer 6: Thanks to the reviewer for his comment. In this regard, a new section called procedure was integrated into the text where the entire data collection process is recorded. (Pages 4, lines 161-173)
Question 7: Please speak in detail about the Hayes macro method in the method section.
Answer 7: Dear reviewer, we appreciate your comment. In this regard, the authors discuss Hayes Macro in section 2.4 [Data analysis] (lines 181-193). However, if you would like us to add any additional information, please let us know.
Question 8: Wouldn't it be better if you used objective methods of measuring physical activity? It is suggested to be mentioned in the limitations section.
Answer 8: We appreciate the reviewer's suggestion for improvement. In this regard, a sentence has been added in the method section, which refers to the need to use objective devices for the assessment of physical activity. (Pages 8, lines 275-278)
Question 9: Please search the research literature more carefully. It seems that similar research has been done recently.
Answer 9: We are very grateful for your interesting question. However, to our knowledge, we have not found any similar recent work. The work that may be most similar is that of (Vaquero-Solís et al., 2020). In this regard, this work focuses on relationships based on Pearson correlations and regression analysis with a different theoretical construction where motivation would be the independent variable as a predictor of physical activity, and this in turn would predict emotional intelligence. However, in the present study we postulate emotional intelligence as a trait associated with physical activity through different types of motivational regulations.
Likewise, as mentioned in the theoretical framework, the work of Porter (2014) is similar to ours since she proposes a mediation analysis based solely on intrinsic regulation as a mediating variable of the association of emotional intelligence with physical activity. Another work of a similar nature is that of sukys et al. (2021) in a population of sports athletes.
Finally, to ensure that there were no other articles that might conflict with ours, we performed a basic search in Web of Science and Schoolar where we were able to identify two papers from the year 2022 (Failde Garrido et al., 2022; and Méndez-Giménez et al., 2022). The first one based on university students and the second one on a sports education model.
To conclude, we would like to point out that if the reviewer refers to any other work that we have not cited, we would like to know about it.
Failde-Garrido, J. M., Ruiz Soriano, L., & Simon, M. A. (2022). Levels of physical activity and their relationship with motivational determinants, self-regulation, and other health-related parameters in university students. Psychological Reports, 125(4), 1874-1895.
Méndez-Giménez, A., del Pilar Mahedero-Navarrete, M., Puente-Maxera, F., & de Ojeda, D. M. (2022). Effects of the Sport Education model on adolescents’ motivational, emotional, and well-being dimensions during a school year. European Physical Education Review, 28(2), 380-396.

Round 2
Reviewer 1 Report
I did have a few minutes to take a look at the manuscript.. The authors have improved (clarified) their intent,
However, English usage continues to be a problem; see attached in just the Abstract. It would take me way too much time to
correct. As I noted, the content is somewhat interesting, but average.
